# Synthesis and Mechanism of a Green Scale and Corrosion Inhibitor

**DOI:** 10.3390/ijms251810150

**Published:** 2024-09-21

**Authors:** Linlin Zhao, Yu Han, Xiaojuan Zhang, Zhongyan Cao, Xiaowei Zhao, Yuxia Wang, Yonghong Cai, Yufeng Wu, Ying Xu

**Affiliations:** 1College of Chemistry and Molecular Sciences, Henan University, Kaifeng 475004, China; zhaolin@henu.edu.cn (L.Z.); 104753221132@henu.edu.cn (Y.H.); zxjnmr@126.com (X.Z.); zycao@henu.edu.cn (Z.C.); yzszxw@henu.edu.cn (X.Z.); wangyuxia@henu.edu.cn (Y.W.); caiyonghong@henu.edu.cn (Y.C.); 2College of Pharmacy and Chemical Engineering, Zhengzhou University of Industrial Technology, Zhengzhou 451150, China

**Keywords:** polyaspartic acid, 5-aminovalerate, scale inhibition, corrosion inhibition, mechanism

## Abstract

A new green water treatment agent, a poly(aspartic acid)-modified polymer (PASP/5–AVA), was synthesized using polysuccinimide and 5-aminovaleric acid (5-AVA) in a hybrid system. The structure was characterized, and the scale and corrosion inhibition performance were carried out with standard static scale inhibition and electrochemical methods, respectively. The mechanism was explored using XRD, XPS, SEM, and quantum chemistry calculations. The results indicated that PASP/5–AVA exhibited better scale and corrosion inhibition performance than PASP and maintained efficacy and thermal stability of the scale inhibition effect for a long time. Mechanistic studies indicated that PASP/5–AVA interferes with the normal generation of CaCO_3_ and CaSO_4_ scales through lattice distortion and dispersion, respectively; the combined effect of an alkaline environment and terminal electron-withdrawing -COOH groups can induce the stable C^-^ ionic state formation in -CH_2_- of the extended side chain, thus enhancing its chelating ability for Ca^2+^ ions. At the same time, the extension of the side chain length also enhances the adsorption ability of the agent on the metal surface, forming a thick film and delaying the corrosion of the metal surface. This study provides the necessary theoretical reference for the design of green scale and corrosion agents.

## 1. Introduction

Global water resources are becoming increasingly strained. Industrial water has become the fastest-growing sector in terms of water consumption while circulating water accounts for two-thirds of the total industrial water consumption [1]. Many reaction processes in chemical and other industries are exothermic during production, and the heat generated during production must be removed in a timely manner to maintain normal production operations. In the industrial recirculating cooling water system, the system is usually required to maintain the operating temperature at 15 °C~45 °C, although the temperature is often high due to various factors. In the long run, the water will continue to evaporate, the concentration of Ca^2+^ and Mg^2+^ ions in the water will gradually increase, and inorganic salts and other ions can easily combine to form limescale. At present, scaling is a serious problem faced by many industries, such as water treatment. Scale deposition on the surfaces of equipment and pipelines, particularly around pipe bends, blocks the equipment and pipelines, reduces the heat transfer efficiency, and causes heat accumulation, which results in accidents [2,3]. In addition, scale deposits cause localized current differentials in pipelines, accelerating the corrosion of equipment and pipelines, which in turn increases operating costs and risk factors [4,5]. Various methods have been used in industry to prevent inorganic scale formation and deposition, but the use of water treatment agents is currently the most effective way [6,7].

There are many industrial water treatment agents used; however, there are still various problems. Chitosan, tannin, cellulose, and other natural polymer scale inhibitors have obvious scale inhibition but also have disadvantages such as high dosages, easy decomposition at high temperatures, and poor scale inhibition [8,9]. Some phosphorus-containing scale inhibitors, for example, organic phosphonates, have better scale inhibition efficiencies; however, they cause eutrophication of water bodies, which hinders the widespread popularization of phosphorus-containing scale inhibitors [10,11]. To solve these kinds of problems, it is urgent to explore and design new water treatment agents that are nonphosphorus, multifunctional, biodegradable, and efficient.

Polyaspartic acid (PASP), with good biodegradability, is an environmentally friendly water treatment agent resulting from the absence of phosphorus and other nutrients in the molecular composition, water solubility, easy biodegradability, and a structure containing many carboxyl groups that chelate Ca^2+^ ions [12,13]. The literature has suggested that PASP exhibits excellent CaSO_4_ scale inhibition, whereas it is less effective in inhibiting CaCO_3_ scale and providing corrosion resistance, so it does not meet the needs of industrial application [14,15]. To further improve the overall water treatment performance of PASP, functional groups or small molecules, such as hydroxyl, carboxylic, heterocyclic, sulfonic groups, etc., are usually introduced into PASP to obtain modified PASP polymers [16,17,18]. Although much has been reported about the structure–activity relationship of water treatment agents, deeper scientific evidence is needed to confirm it [19,20].

To reveal the structure–performance relationships of PASP, the effect of side chain length on the scale and corrosion inhibition performance of PASP was investigated. A new PASP derivative (PASP/5–AVA) was synthesized using nucleophilic compounds as ring openers to break the amide bond in poly (succinimide) and by introducing the 5-aminovaleric acid (5−AVA) functional group into the PASP side chain. The structure of 5-AVA was characterized by ^1^H NMR and FT-IR. The scale inhibition performance, including concentration, the solution pH value, and reaction temperature, was investigated according to the static scale inhibition method. In addition, the corrosion inhibition performance of 5−AVA was also tested by electrochemical methods. Finally, the scale and corrosion inhibition mechanism was explored by optimizing its structure using quantum chemical calculations.

## 2. Results and Discussion

### 2.1. Structural Characterization of PASP/5–AVA

#### 2.1.1. The Results Analysis of FT-IR and ^1^H-NMR

The FT-IR spectra of PASP and PASP/5–AVA are shown in Figure 1a. From the FT-IR spectra of PASP, the peak at ~1620 cm^−1^ is a stretching vibration absorption peak of the amide C=O bonds. The C-N stretching vibration peak at ~1400 cm^−1^ overlapped with the N-H bending vibration absorption peak. In addition, the stretching vibration peak of the amide N-H bonds in PASP is shown at ~3500 cm^−1^. The -OH bending vibration of the carboxyl groups appeared at 950–900 cm^−1^. The absorption peaks at ~2930 cm^−1^ are attributed to the stretching vibrations of the -CH_2_- group in the main chain, and the stretching vibrations of the -CH- groups resulted in the peaks at ~ 2890 cm^−1^. This indicated that PASP was obtained from the PSI ring opening reaction. In the FTIR spectrum of PASP/5–AVA, besides the peaks for PASP, the new peaks located at ~1401 cm^−1^, ~1629 cm^−1^, and ~3393 cm^−1^ were assigned to stretching vibrations of the C-N, C-O, and N-H bonds in the amides of PASP/5–AVA, respectively. Additionally, in the PASP/5–AVA polymer side chains, the O-H bonds stretching vibrations peak in the -COOH group appeared at ~3400 cm^−1^, while the asymmetric stretching vibrations of the -CH_2_- groups appeared at ~3000 cm^−1^, and the two peaks coincided with the stretching vibrations of the N-H. However, the O-H stretching vibration gave a broad and strong peak due to the association of the hydroxyl compounds. In addition, the absorption peak at ~2846 cm^−1^ was the symmetric stretching vibration of -CH_2_- in the PASP/5AC polymer side chain. Thus, the PASP/5–AVA graft copolymer was synthesized.

Figure 1b shows the ^1^H-NMR spectra of PASP and PASP/5–AVA polymers in D_2_O. The -CH- and -CH_2_- groups in PASP displayed broad peaks at 4.45 and 2.65 ppm (① and ②, with an integrated ratio of 1:2), respectively (Figure 1b). In the spectrum of PASP/5–AVA, it can be seen that the signal peaks of the -CH_2_- (⑥ and ⑦) and -CH_2_- (⑧) groups are located at δ_H_ = 1.30 and δ_H_ = 2.11 ppm, respectively. It is worth noting that -CH_2_- groups (⑥ and ⑦) in the side chain did not show coupling due to the symmetry of the molecule. Broad peaks at 4.43 and 2.62 ppm (③ and ④) with chemical shifts similar to those of PASP appeared in the spectrum. Meanwhile, the broad triplet at 3.0 ppm (⑤) was assigned to -NH-CH_2_- groups in the side chains of PASP/5–AVA. Combined with the infrared spectrogram results, these NMR data confirmed that PASP/5–AVA was synthesized successfully.

#### 2.1.2. Gel Chromatography (GPC) Analysis

Polymers are usually composed of macromolecular homologs with different molecular weights [21]. The molecular weight polydispersity dispersity (*Ð*, *M_w_*/*M_n_*) is one of the important indices used to characterize a polymer, which affects the stability of the polymer, the temperature resistance, etc. Generally, a narrow molecular weight distribution corresponds to a stable polymer [22]. Gel permeation chromatography (GPC) was used to measure the molecular weights of PASP and PASP/5–AVA to clarify the characteristics of PASP/5–AVA, and the results are displayed in Table 1. From the results, the weight-average molecular weight (*M_w_*) of the prepared PASP/5–AVA was 14,109 Da, and the number-average molecular weight (*M_n_*) was 11,328 Da; thus, the *Ð* was 1.245. It is noteworthy that the actual *M_n_* of PASP/5–AVA obtained by the GPC method is much higher than its theoretical value. This result indicates that the PASP/5–AVA polymer is not composed of single chain segments, but there are longer chain segments, i.e., there are multiple chain segments with different molecular weights in the PASP/5–AVA polymer, and therefore, the actual *M_n_* in the GPC analysis result is much larger than the theoretical value.

The literature shows that the molecular weights of polymers are closely related to their scale inhibition capabilities; in general, molecular weights within a certain range indicate good inhibition efficiency [23]. The excessive molecular weight leads to the encapsulation of some of the carboxyl groups and other functional groups in the molecule, which reduces its scale and corrosion inhibition ability [24]. The above data confirm that PASP and PASP/5–AVA have narrower molecular weight distributions, suggesting that PASP/5–AVA polymers are well dispersed, highly stable, and potentially have good scale resistance.

#### 2.1.3. Zeta Potential Analysis

The zeta potential shows the polymer charge and is a measure of the mutual repulsions or attractions of particles. To confirm the charges, the zeta potentials of PASP and PASP/5–AVA were determined as −4.77 and −8.12, respectively, indicating that both carried negative charges [25]. The charge of PASP/5–AVA would attract positively charged Ca^2+^ ions, thereby inhibiting the combination of Ca^2+^ ions with CO_3_^2−^ and SO_4_^2−^ in circulating water and reducing scale deposition. On the other hand, the potential of PASP/5–AVA was higher than that of PASP, indicating that PASP/5–AVA adsorbs more effectively on the surfaces of scale crystals and neutralizes the charge. The electrostatic repulsion between similar charges disperses scale crystals in circulating water and prevents deposition of the scale. Therefore, PASP/5–AVA is potentially a good scale inhibitor.

### 2.2. Scale Inhibition Results Analysis of PASP/5–AVA

#### 2.2.1. Analysis of CaCO_3_ Scale Results

To explore the CaCO_3_ scale inhibition of PASP/5–AVA, the relationship between the inhibitor concentration, the environmental temperature and action time, and the scale inhibition efficiency were measured, and the results are listed in Figure 2. Figure 2a reveals the effects of scale inhibitor concentrations on CaCO_3_ scaling. From Figure 2a, it can be clearly seen that with the gradual increase in concentration, the scale inhibition efficiency of PASP and PASP/5–AVA on CaCO_3_ showed a trend of gradual improvement up to the concentration of 40 mg/L when the scale inhibition efficiency of both reached the maximum value. It should be noted that over the experimental concentration range, the scale inhibition efficiency of PASP on CaCO_3_ was always lower than that of PASP/5–AVA. It is exciting that when the concentrations were 5 mg/L, compared with PASP, the scale inhibition efficiency of PASP/5–AVA was increased by about 45%. At a dose of 10 mg/L, the CaCO_3_ scale inhibition efficiency of PASP/5–AVA was about 27% higher than that of PASP. When the concentration exceeded 30 mg/L, the CaCO_3_ scale resistance efficiency of PASP/5–AVA remained nearly constant and was stable at approximately 65.32%. However, after the concentration exceeded 43 mg/L, the scale inhibition efficiency declined slightly. As the length of the side chains increased, the steric hindrance of the polymer decreased, and the coordination sites for the chelated Ca^2+^ ions increased, thus chelating more Ca^2+^ ions and hindering the deposition of CaCO_3_ scale, which had a better scale inhibition ability at low concentrations. When the concentration of the polymer was too high, the chains were intertwined, and functional groups such as -COOH were wrapped, which reduced the probability of binding the Ca^2+^ ions and decreased the scale inhibition efficiency.

Moreover, when the concentration was 30 mg/L, the effect of temperature on PASP/5–AVA against CaCO_3_ scale was investigated, and the results are listed in Figure 2b. In the experimental temperature range of 40 °C~80 °C, the anti-CaCO_3_ scale efficiency of PASP/5–AVA was still higher than that of PASP. It is noteworthy that when the temperature was increased to 80 °C, the anti-CaCO_3_ scale efficiency of PASP/5–AVA still reached 60%, while PASP was only 48.9%. The above results indicated that with the increase in temperature, the inhibition efficiency of both PASP and PASP/5–AVA on the CaCO_3_ scale showed a decreasing trend. However, in the whole range of test temperatures, the scale inhibition efficiency of PASP/5–AVA on CaCO_3_ was higher than that of PASP and was still maintained at about 60% even at 80 °C, which fully demonstrated the inhibition of CaCO_3_ scale by PASP/5–AVA. This shows that PASP/5–AVA has high thermal stability for CaCO_3_ scale inhibition. At the same time, this result also indicates that the extension of the side chain can not only improve the scale-inhibiting efficiency of PASP but also improve the thermal stability of its scale-inhibiting effect. The reason for this is that the -CH_2_- on the side chain is acidic due to the carbonyl group in the adjacent -COOH position, and in the alkaline test environment, the -CH_2_- on the side chain forms negative Ca^2+^ ions, so it can also interact with positively charged Ca^2+^ ions, adsorbing more Ca^2+^ ions, and at the same time improving the thermal stability of the deterrent’s action.

In addition, to determine the effect of heating time on the anti-CaCO_3_ efficiency, the scale inhibition efficiency of PASP and PASP/5–AVA was observed at the dosage of 30 mg/L, which is shown in Figure 2c. The anti-CaCO_3_ scale efficiencies of PASP and PASP/5–AVA remained almost stable as the heating time was increased. After 12 h of heating, the anti-CaCO_3_ scale efficiency of PASP/5–AVA remained at approximately 60%, whereas that of PASP was approximately 50%. In addition, the anti-CaCO_3_ scale efficiency of PASP/5–AVA was always higher than that of PASP. In addition to the chelating effect of -COOH at the end of the side chain, -CH_2_- on the side chain is negatively charged by -COOH, which allows better adsorption of Ca^2+^ ions. With time, the adsorption sites of PASP/5–AVA were gradually occupied, resulting in a decreasing trend of scale inhibition. However, in the tested heating time range, compared with PASP, the persistence of the anti-CaCO_3_ scale inhibiting effect of PASP/5–AVA was significantly higher.

#### 2.2.2. Analysis of CaSO_4_ Scale Results

The crystallization kinetics of CaSO_4_ are different from those of CaCO_3_, so the optimal concentration and inhibition efficiency may differ for the same agent used in inhibiting CaCO_3_ scale and CaSO_4_ scale [26]. To explore the anti-CaSO_4_ scale effects of inhibitors, the effects of the scale inhibitor concentration, the experimental temperature, and the heating time against the CaSO_4_ scale were determined. Figure 3a demonstrates the anti-CaSO_4_ scale effects of PASP and PASP/5–AVA. From the results, the anti-CaSO_4_ scale efficiencies of both PASP and PASP/5–AVA increased with increasing concentrations. Similar to CaCO_3_, the anti-CaSO_4_ scale efficiency of PASP/5–AVA was better than PASP over the range of experimental concentrations. At the dosage of 2 mg/L, the anti-CaSO_4_ scale efficiency of PASP/5–AVA was approximately 80%, but PASP showed only 20% efficiency. When the scale inhibitors concentrations were 3 mg/L, the anti-CaSO_4_ scale efficiency of PASP/5–AVA, which was close to 100%, was 69% higher in comparison to PASP. When the concentrations exceeded 3 mg/L, the anti-CaSO_4_ scale rate of PASP/5–AVA was stable at 100%. However, when the dosage was 6 mg/L, the anti-CaSO_4_ scale efficiency of PASP was close to 100%. These data show that PASP/5–AVA is also highly effective in inhibiting CaSO_4_ scale as well as CaCO_3_ scale.

The effect of temperature on scale inhibitors against CaSO_4_ scale was observed with the concentration of 3 mg/L, and the relationship is shown in Figure 3b. With a gradual increase in the experimental temperature, the anti-CaSO_4_ scale efficiency of PASP showed a decreasing trend, however, the scale inhibition rate of PASP/5–AVA on the CaSO_4_ scale basically remained unchanged. When the experimental temperature was less than 85 °C, the anti-CaSO_4_ scale efficiency of PASP/5–AVA was basically maintained at a level of about 99%; at 90 °C, the anti-CaSO_4_ scale rate was still 81%. On the contrary, the scale inhibition efficiency of PASP on the CaSO_4_ scale decreased by 60.84% at 80 °C; when the temperature was raised to 90 °C, the anti-CaSO_4_ scale rate of PASP on the CaSO_4_ scale was only 20.14%. These results indicate that the extension of the side chain can not only effectively improve the scale inhibition ability of PASP against CaSO_4_ scale but also improve the heat resistance of its scale inhibition ability in a certain temperature range, which in turn expands its range of use in industrial applications.

Figure 3c displays the relationship between the experimental time and CaSO_4_ scale resistance when the scale inhibitor was 3 mg/L. The scale inhibition rate of PASP/5–AVA on the CaSO_4_ scale remained essentially unchanged over the test time range, and even when the static experiment lasted for 24 h, its scale inhibition efficiency was still maintained at around 100%. On the other hand, the scale inhibition rate of PASP against the CaSO_4_ scale began to gradually decrease when the test time exceeded 10 h, and the anti-CaSO_4_ scale efficiency decreased to 62% when the time lasted 24 h. This result confirms the speculation of the above results that the extension of the side chain not only promotes the adsorption of more Ca^2+^ ions by the PASP scale inhibitor but also the formation of a more stable structure which prolongs the duration of the scale inhibitor effective time.

### 2.3. Calcium Scale Analysis

#### 2.3.1. SEM Analysis of Calcium Scales

To investigate the inhibition mechanism of PASP/5–AVA, the morphology of CaCO_3_ and CaSO_4_ scales was scanned with and without inhibitors by scanning electron microscopy (SEM), and the images are displayed in Figure 4a–c. It was clearly seen that when the scale inhibitor was absent, the CaCO_3_ scale resembled calcite, with regular lozenge shapes, a regular structure, and smooth surfaces, which was the most stable crystalline form. When the concentration of PASP was 30 mg/L, the crystals of the CaCO_3_ scale began to change, the original regular cubic calcite shape disappeared, and multiple crystal shapes were formed, while the surface was no longer smooth and became rough with a fish scale appearance, and the original calcite was destroyed (Figure 4b). After adding 30 mg/L PASP/5–AVA, the original regular calcite morphology of the CaCO_3_ scale crystals was more seriously deformed, and the overall size of the crystals was smaller; the surface showed a loose bloom, like light cotton, and was easily washed away (Figure 4c). The above results indicate that both PASP and PASP/5–AVA are involved in the formation of CaCO_3_ scale crystals and can effectively destroy the formation of CaCO_3_ crystals, i.e., effective lattice distortion; it can also be seen that the extension of the side chain can effectively increase the destructive ability of PASP on the formation of CaCO_3_ scales and further reduce the scale deposition. Indirectly, the above results again confirmed that the extension of the side chain induced PASP to interact with more Ca^2+^ ions and, thus, improve the scale inhibition efficiency of PASP.

Similarly, the CaSO_4_ deposits showed needle-like particles with regular structures and smooth surfaces when the scale inhibitor was absent (Figure 4e). At the dosage of 3 mg/L PASP, the original regular shapes of the CaSO_4_ crystals were damaged, and the obvious angles disappeared, showing a jagged shape, and the surface was also damaged to some extent, forming a scaly morphology (Figure 4e). When 3 mg/L PASP/5–AVA was added, the structure of the CaSO_4_ crystals was severely damaged; at the same time, the whole surface of the crystals appeared very rough, showing a lamellar and flowered morphology, and became loose (Figure 4f). When crystals crystallize rapidly, the number of nuclei increases, forming fine, needle-like, or dendritic crystals. Conversely, if the crystallization rate is too slow, loose and thick crystals are formed. It can be seen that PASP/5–AVA inhibits the formation of the CaSO_4_ scale more than PASP. This result also suggests that the extension of the side chain length effectively promotes the ability of PASP to inhibit CaSO_4_ crystal formation. The destruction of CaSO_4_ crystals leads to the loosening of the calcium scale in solution and a reduction in the amount of deposition.

#### 2.3.2. XRD Analysis of Calcium Scales

To investigate the action mechanism of the scale inhibitors, the CaCO_3_ and CaSO_4_ crystals were studied using XRD to monitor their crystalline phase changes. The XRD spectrums of CaCO_3_ and CaSO_4_ in the presence and absence of the scale inhibitor are shown in Figure 5. As seen in Figure 5a, the peaks for CaSO_4_ were located at 12.04°, 21.78°, 23.45°, and 29.69° in the absence of inhibitor. The peak intensity was high and sharp, indicating higher crystallinity and larger particles. The XRD diffraction peak positions of CaSO_4_ did not change significantly when PASP or PASP/5–AVA was added. However, compared to the diffraction peaks without the inhibitor, whether the PASP or PASP/5–AVA was added, it would result in a decrease in peak height and some broadening, especially for PASP/5–AVA. This suggests that the addition of PASP reduces the size and crystallinity of the grains and that the extension of the side chains increases the potency of PASP. Combined with the SEM results of the CaSO_4_ scales, this further suggests that PASP/5–AVA does not alter the internal structure of CaSO_4_ scale crystals but merely hinders crystal growth by adsorbing more Ca^2+^ ions.

The data of XRD peaks for aragonite are usually seen at 26.13°, 27.13°, 33.03°, 37.18°, 37.78°, 38.33°, 41.07°, 42.76°, 45.78°, 48.29°, 50.11°, 52.35°, and 52.81°; the peaks at 22.98°, 29.28°, 35.86°, 39.33°, 47.42°, and 57.29° corresponded to calcite, and the diffraction peaks at 21.02°, 24.94°, 27.05°, 32.82°, 43.89°, 49.15°, 50.13°, and 55.84° corresponded to vaterite [27]. Figure 5b displays the XRD patterns for CaCO_3_ with and without the antiscaling agent. When the scale inhibitor was absent, diffraction peaks can be clearly seen mainly at about 29.9°, 30°, 35.8°, 39.3°, 47.2°, and 53°, which are typical diffraction characteristic peaks of calcite. However, when the PASP or PASP/5–AVA was added, a clear shift of the original typical calcite diffraction peaks could be clearly observed, and the change in shift varies between the two. After the addition of PASP, compared without scale inhibitor, new diffraction peaks at about 26°, 27.1°, 33.56°, 41.12°, 45.5°, and 48.2° are presented, which are typical characteristic diffraction peaks of aragonite. When the PASP/5–AVA was added, similar to those of PASP, the diffraction peaks were at 26°, 27.1°, and 33.56°, except that the characteristic diffraction peaks of calcite at about 57° almost disappeared. This result fully demonstrates that the PASP inhibitor resulted in the transformation of some CaCO_3_ crystals from the calcite form to the aragonite. The extension of side chains resulted in the disappearance of the calcite of CaCO_3_ and basically its complete transformation to the aragonite, which is an unstable form of CaCO_3_. In addition, consistent with the SEM results, the addition of PASP altered the structure of the calcium crystal by adsorption of Ca^2+^ ions and other effects, thereby disrupting the normal formation of calcium scale, i.e., the lattice distortion effect. The side chain lengthening of PASP increased its ability to distort the lattice, resulting in the formation of smaller, less crystalline particles on the CaCO_3_ scale.

#### 2.3.3. XPS Analysis of Calcium Scales

To further explore the action mechanism of the PASP/5–AVA inhibitor, the CaCO_3_ and CaSO_4_ scales formed in the static scale inhibition tests were observed using X-ray photoelectron spectroscopy (XPS). Figure 6 shows the high-resolution Ca 2p XPS data of calcium scales formed in the presence and absence of scale inhibitors.

When the scale inhibitor was absent, the peaks at 347.62 and 343.94 eV were Ca 2p_1/2_ and Ca 2p_3/2_ peaks of the CaCO_3_ scale, respectively (Figure 6a–c). With 30 mg/L PASP in the solution, the binding energies of Ca 2p_1/2_ and Ca 2p_3/2_ were 0.43 eV and 0.3 eV lower than that without the scale inhibitor, respectively. When 30 mg/L of PASP/5–AVA was added, the binding energies of Ca 2p_1/2_ and Ca 2p_3/2_ decreased by 0.53 eV and 0.46 eV, respectively. The binding energies change of Ca 2p_1/2_ and Ca 2p_3/2_ suggested that PASP or PASP/5–AVA changed the chemical environment of Ca^2+^ ions in the CaCO_3_ scale; the decrease of binding energy further indicated the formation rate of CaCO_3_ crystals was slowed down, thus reducing the formation of CaCO_3_ scales. Lengthening of the side chains increased the lattice distortion ability of PASP on the CaCO_3_ scale, resulting in significant displacement.

Similar to the XPS results for the CaCO_3_ scale, after the addition of PASP or PASP/5–AVA, the Ca 2p peaks shift for the CaSO_4_ scale also changed (Figure 6e,f), and the binding energies of Ca 2p_1/2_ were increased by 0.16 eV and 0.28 eV, respectively. The binding energies of the Ca 2p_3/2_ also changed by 0.18 eV and 0.26 eV, respectively. These changes confirm that both the electronegative groups in PASP and PASP/5–AVA could act on Ca^2+^ ions, thus hindering the normal formation of the CaSO_4_ scale. In addition, due to the long length of the PASP/5–AVA side chain and the action of the alkaline environment and -COOH groups, it has a stable C^−^ property in -CH_2_- near -COOH on the side chain, so it could act on more Ca^2+^ and have a more stable binding capacity than PASP. The above results suggested that the PASP and PASP/5–AVA could bind to Ca^2+^, and the Ca atoms lose more electrons, changing the original chemical environment and shifting the peak to higher energies. The extent of the shift suggests that the lengthening of the side chain may encourage PASP to bind more Ca^2+^ ions, thereby preventing the growth of CaSO_4_ scale crystals. However, it should be noted that the inhibition mechanism of PASP/5–AVA on the CaSO_4_ scale is not due to the lattice distortion effect. Summarizing the SEM and XRD results, it is thought that the scale inhibitor inhibited the growth of CaSO_4_ crystals mainly through the dispersion effect.

#### 2.3.4. Surface Energy Analysis of CaCO_3_ Scale

Furthermore, to further explore the mechanism of PASP/5–AVA, the pH changes in solution during CaCO_3_ scale growth by PASP and PASP/5–AVA were tested with a pH meter. Specifically, to determine the induction time (*t*_ind_) of CaCO_3_ scale formation, the pH changes of the solution were monitored at four different supersaturation ratios (S) of CaCO_3_ solutions with and without the scale inhibitor. The concentrations of CaCO_3_ in the four solutions (A, B, C, and D) were 0.015 mol/L, 0.02 mol/L, 0.025 mol/L, and 0.03 mol/L, respectively. The relationship curves are described in Figure 7. Table 2 shows the *t*_ind_ for solutions without scale inhibitors and with different dosages of scale inhibitors (Figure 7a).

Combined with Table 2 and Figure 7, it can be seen that a reasonable extension of the *t*_ind_ occurred for all four CaCO_3_ solutions with the dosage of 4 mg/L PASP; however, a significant extension of the *t*_ind_ occurred for all four solutions after the addition of the same concentration of PASP/5–AVA compared to the absence of the inhibitor (Figure 7b,c). The maximum *t*_ind_ after adding PASP/5–AVA was 53.71 min in solution A, while the *t*_ind_ of the blank solution was only 1.50 min, and the induction time for the solution with an appropriate amount of PASP was 2.25 min. In addition, the surface energy of the calcium scale was calculated according to Equation (4) (Figure 7d), and the crystal surface energies of the solution without the scale inhibitor and the solutions with PASP and PASP/5–AVA were 39.6 mJ·m^−2^, 45.1 mJ·m^−2^, and 63.9 mJ·m^−2^, respectively. These results clearly indicated that the extension of the side chain could induce PASP to better chelate more Ca^2+^ ions, reducing the local supersaturation of the solution, inhibiting crystal nucleation, and making crystallization more difficult, and this conclusion was also confirmed by the change in surface energy. However, as the Ca^2+^ concentration increased, the site of action of the agent was occupied, leading to a gradual shortening of the *t*_ind_, which also confirms that the ion concentration ratio also affects the *t*_ind_.

### 2.4. Corrosion Inhibition Analysis of PASP/5–AVA

#### 2.4.1. Analysis of Electrochemical Polarization Curves

Analyses of polarization curves are used for studying metal corrosion to reveal the metal corrosion mechanism [28]. Therefore, the potentiodynamic polarization changes of mild steel samples with and without inhibitors were measured at room temperature. To obtain a stable open-circuit potential, the carbon steel electrode was immersed in a 3.5% NaCl solution for 1 h before being used for measurements. The polarization curves results are displayed in Table 3 and Figure 8, from which potentiodynamic constants such as corrosion potential (*E_corr_*), the cathodic and anodic Tafel slopes (*β*_c_ and *β*_a_), surface coverage (*θ*), corrosion inhibition efficiency (*η*_p_,%), and corrosion current (*I_corr_*) of corrosion inhibitors with various concentrations were obtained.

When the inhibitors were absent, the corrosion current density (*I_corr_*) and the corrosion potential (−*E_corr_*) were 3.516 mA/cm^2^ and 0.6775 V vs. SCE, respectively. These data indicate a relatively serious corrosion trend of the metal and easy corrosion. After adding the inhibitor, with the increase of the agent concentration, the corrosion potential (*E_corr_*) moved in the positive direction, and the self-corrosion potential decreased, indicating a corrosion tendency weakening of the metal surface. Similarly, the corrosion current density (*I_corr_*) was also gradually reduced, indicating that anode corrosion was suppressed. As the concentration of the agent increases, the polarization curve of the self-corrosion potential gradually shifts to positive; the *I_corr_* gradually decreases, and the corrosion inhibition performance is better. In addition, Figure 8 shows that both PASP and PASP/5–AVA caused a large change in the anodic Tafel slope (*β*_a_), while the cathodic Tafel slope (*β*_c_) did not change significantly. This result indicated that PASP and PASP/5–AVA inhibited the corrosion of the metal by altering the anodic corrosion process and were typical anodic corrosion inhibitors. However, compared with PASP, the addition of PASP/5–AVA resulted in a greater positive shift of the corrosion potential and a smaller corrosion current, and the corrosion inhibition efficiency and surface coverage of PASP/5–AVA were also relatively improved. At 100 mg/L, compared with PASP, the corrosion inhibition efficiency and the surface coverage of PASP/5–AVA increased by 29.16% and 0.29, respectively. This result fully demonstrates that the extension of the side chains can improve the adsorption capacity between PASP and the surface of carbon steel and, thus, effectively protect the metal surface to avoid corrosion.

#### 2.4.2. Analysis of Electrochemical AC Impedance Spectra

To investigate the corrosion inhibition performance and mechanism of corrosion inhibitors, electrochemical impedance spectroscopy (EIS) was used [29,30]. The electrochemical impedance profiles were obtained with and without corrosion inhibitors, and the results are shown in Figure 9. The relevant parameters are exhibited in Table 4.

From Figure 9 and Table 4, it can be seen that the impedance spectra with and without corrosion inhibitor exhibit the shape of a single capacitive resistive arc. It is indicated that the dissolution of 20^#^ carbon steel is mainly controlled by charge transfer. At the same time, compared to without corrosion inhibitor, the addition of PASP or PASP/5–AVA increases the high-frequency resistive arc radius, and with increasing corrosion inhibitor concentration, the high-frequency resistive arc radius increases. It is worth noting that the high-frequency capacitive resistance arc radius of PASP/5–AVA added at the same concentration is significantly greater than that of PASP. From Table 4, the inhibition efficiency of PASP/5–AVA is increased by 16.73% compared with PASP at the dosage of 30 mg/L; at 70 mg/L, the corrosion inhibition efficiency is increased by 24.2%. The above results indicate that the presence of corrosion inhibitors leads to corrosion occurring, reaction transfer resistance increases, and double-layer capacitance is significantly reduced, thus slowing down the corrosion rate of the metal. With the increase of inhibitor concentration, the charge transfer resistance gradually becomes larger, and the metal corrosion rate is gradually reduced. It can also be seen from the impedance spectra of the low-frequency part in Figure 9, a diagonal line with an inclination of 45 °C. This indicates that a thicker and denser passivation film was formed on the electrode surface after adding corrosion inhibitors, resulting in a large increase in film resistance; the migration process of ions is greatly inhibited, effectively preventing the corrosion of the metal. Consistent with the above results of the electropolarisation curves, the extension of the side chain is conducive to improving the adsorption capacity between PASP and the metal surface. The solution resistance increased, and the high-frequency capacitive resistance presented an increase, which reduced the corrosion rate of the metal to a greater extent and improved the corrosion inhibition rate.

### 2.5. Action Mechanism of PASP/5–AVA

#### 2.5.1. Mechanism Analysis of Scale Inhibition

To explore the scale inhibition mechanism, Gaussian quantum chemical calculations were performed on PASP/5–AVA to obtain its optimal structure and charges, and the results are shown in Figure 10. From the data in Figure 10, the N (−0.668848, −0.57034), O (−0.37315, −0.39235, −0.42867, −0.56595), and C (−0.2969, −0.29006, −0.24382, −0.25541) atoms of the PASP/5–AVA structure carry a large number of negative charges, providing more ideal binding sites for Ca^2+^ ions. Consequently, when PASP/5–AVA is dispersed in circulating water, it can combine with more of the positively charged metal ions, thereby intervening in crystal formation and reducing the formation of CaCO_3_ and CaSO_4_ scales. Moreover, Combining the SEM, XRD, and XPS results of calcium scales, it can be concluded that the chelating number and chelating capacity of PASP/5–AVA for Ca^2+^ ions is much higher than PASP. The above results imply that PASP/5–AVA interferes mainly with the normal formation of CaCO_3_ and CaSO_4_ through lattice distortion and dispersion effects, respectively. At the same time, the side chain extension improves the scale inhibition capacity of PASP. Analyzing the reasons, as industrial circulating water is often alkaline, the -CH_2_- on the extended side chains of PASP/5–AVA is prone to form a carbon negative ion which is terminally attached to this electron-absorbing -COOH group, resulting in a more stable structure. The -COOH and the stable negatively charged -CH_2_- group carried by PASP/5–AVA interact with the positively charged Ca^2+^ in the solution, thus improving its scale inhibition efficiency. The side chain extension of PASP/5–AVA also reduces the intramolecular -COOH interactions compared to PASP, thereby increasing its stability against Ca^2+^ ion chelation. Thus, it can be seen that proper extension of the side chain can enhance the scale inhibition ability of PASP.

#### 2.5.2. Analysis of Corrosion Inhibition Mechanism

To investigate the anti-corrosion mechanism of PASP/5–AVA, the molecular orbital density distributions of PASP and PASP/5–AVA were optimized using density functional theory. The results are listed in Figure 11, and the corresponding parameters are given in Table 5. The data indicate that the electron density of PASP and PASP/5–AVA is concentrated on highly negatively charged N and O heteroatoms. According to quantum chemical theory, the corrosion inhibition performance of the inhibitors was closely related to the energies and the energy gap (Δ*E*) of the highest and the lowest unoccupied molecular orbital (*E*_HOMO_ and *E*_LUMO_). A high *E*_HOMO_ indicates that the molecule readily donates electrons to suitable low-energy acceptors with empty molecular orbitals. The lower the *E*_LUMO_ value, the better the ability of the molecule to accept electrons. Furthermore, the smaller the Δ*E* is, the more likely the molecule is to donate and accept electrons.

The data in Table 6 show that PASP/5–AVA had a higher *E*_HOMO_, a lower *E*_LUMO_, and a smaller Δ*E* compared with PASP, which indicates that PASP/5–AVA is more likely to interact with metals through the N and O atoms. Moreover, the number of electrons transferred (Δ*N*) by PASP/5–AVA and PASP were 0.5633 and 0.4708, respectively. These values mean that the electrons on the molecule of PASP/5–AVA are more likely to be transferred to the metal surface, and the metal surface could be effectively protected and avoid being corroded. The above results clearly indicate that the side chain extends due to the increase of -CH_2_- groups and changes the atom characteristics of the PASP molecule to some extent. This is conducive to the transfer of electrons to the metal surface, thereby improving the stability of the agent adsorbed on the metal surface, forming a protective film to prevent metal corrosion.

## 3. Materials and Methods

### 3.1. Reagents and Equipment

The reagents and equipment used in this experiment are listed in Table 6. Deionized water (DI) was prepared in the laboratory and used as the solvent for rinsing.

### 3.2. Synthesis of PASP and PASP/5–AVA

Synthesis of PASP: 400 mg NaOH (10 mmol) was dissolved in 1 mL H_2_O and set aside. Then, 984 mg PSI (10 mmol) was dissolved in 10 mL distilled water in a 100 mL reaction flask, and the flask was placed in a water bath at 40 °C with stirring for 24 h. During the reaction, the prepared NaOH solution was added dropwise to the reaction flask. After the reaction, PASP was purified by dialysis in distilled water using a dialysis bag with a molecular weight of 1000 kD, and then the water was removed by vacuum distillation to give PASP. The synthetic reaction for PASP is shown in Figure 12a.

Synthesis of PASP/5–AVA: First, a mixture of 0.984 g polysuccinimide (PSI, 10 mM) and 25 mL distilled water were added to a 50 mL round-bottomed flask with magnetic stirring for 6 h, and the temperature was kept at 60 °C. Then, 1.171 g of 5-aminovaleric acid (5-AVA, 10 mM) dissolved in 5 mL N,N-dimethylformamide (DMF) was added to the above solution. The temperature of the system was adjusted to 40 °C. Subsequently, the pH value of the reaction solution was adjusted to 9 with 0.1 mol/L NaOH solution. After 24 h, a yellowish-brown viscous solid was formed. Then, the product was precipitated with 100 mL of anhydrous ethanol. The precipitate was moved to a drying oven and dried at 60 °C for 24 h. To purify the samples, they were dissolved in pure water, purified by a dialysis membrane, and then spun to obtain the PASP/5–AVA solid product (yield, 96%). The synthetic reaction for PASP/5–AVA is illustrated in Figure 12b. The above-mentioned polymer was put into a reserve solution with a certain concentration for the performance evaluation.

### 3.3. Characterization of PASP/5–AVA

The structural characteristics of PASP/5–AVA were performed using ^1^H nuclear magnetic resonance spectroscopy (^1^H NMR; AVANCE 400 MHz NMR spectrometer, Bruker Optics, Bilerika, Germany) and Fourier transform infrared spectroscopy (VERTEX 70 FTIR spectrometer, Bruker Optics, Bilerika, Germany). The gel permeation chromatography (PL-GPC50, Agilent, Palo Alto, USA) and a nanoparticle size and zeta potential analyzer (Nano ZS Malvern Instruments Co., Ltd., Marvin, England) were used for determining the molecular weight and the zeta potential of the PASP/5–AVA, respectively.

### 3.4. Static Scale Inhibition Experiment

The static scale inhibition method (GB/T 16632-2019) was used to assess the scale inhibition effect of PASP/5–AVA on the CaCO_3_ scale [31]. Specifically, a test solution including Ca^2+^ (240 mg/L) and HCO_3_^−^ (732 mg/L) was kept in an 80 °C water bath and, after 10 h, was cooled to room temperature. Next, 0.01 mol/L of disodium ethylenediaminetetraacetate (EDTA-2Na) solution was used to titrate the filtrate to determine the Ca^2+^ ions concentration. Similarly, the scale inhibition for PASP/5–AVA against the CaSO_4_ scale was evaluated according to industry standard test methods (QSY126-2014) [32]. To determine CaSO_4_ scale resistance, a solution including Ca^2+^ (2000 mg/L) and SO_4_^2-^ (480 mg/L) was heated in a 70 °C water bath for 6 h and then cooled to room temperature. The calcium ion concentration was measured as above. The inhibition efficiencies against CaCO_3_ and CaSO_4_ scales (*η*, %) were calculated according to Formula (1).
(1)$$\eta =\frac{C_{2}-C_{1}}{C_{0}-C_{1}}\times 100\%$$

*C*_0_ represents the initial Ca^2+^ concentration of the water samples without scale inhibitor before heating, and *C*_1_ and *C*_2_ represent the Ca^2+^ concentrations of the water samples without and with scale inhibitor after heating, respectively. Each value was the average value of the experiment repeated three times to minimize data errors.

### 3.5. Calcium Scale Characterization

The surface morphologies of the scale samples were observed with field emission scanning electron microscopy (FESEM). X-ray powder diffraction (XRD) and X-ray photoelectron spectroscopy (XPS) were used to investigate the structures and Ca 2p binding energies of CaSO_4_ and CaCO_3_ crystals, respectively.

### 3.6. Determination of Scale Crystal Surface Energy

The formation process of CaCO_3_ is listed in Equation (I). Crystallization of CaCO_3_ results in a decrease in pH value due to the release of H^+^. Therefore, the point at which the pH of the solution dropped significantly was regarded as the induction time (*t_ind_*) for CaCO_3_ crystal nucleation [33,34]. The pH value change of the solution was detected to investigate the effects of PASP and PASP/5–AVA against CaCO_3_ crystals. The specific process is as follows. Four CaCl_2_ and NaHCO_3_ solutions with equal volumes and molar concentrations were mixed to obtain solutions A, B, C, and D, and the concentrations of CaCO_3_ were 0.015 mol/L, 0.02 mol/L, 0.025 mol/L, and 0.03 mol/L, respectively. The corresponding supersaturation ratio (S) of A, B, C, and D were 93.325, 144.524, 204.174, and 269.153, respectively. A certain amount of scale inhibitor was added to the premixed solution. The reaction temperature was kept at room temperature. The surface energy (*γ*) of the CaCO_3_ crystals was obtained according to Formula (2).
Ca^2+^ + HCO_3_^−^→CaCO_3_↓ + H^+^
(2)$$Int_{\mathrm{ind}}=B+\frac{\beta \gamma^{3}v_{m}^{2}N_{A}f(\theta )}{R^{3}T^{3}{\left(LnS\right)}^{2}}$$

*B* and *γ* represent a constant and surface energy (J·m^−2^), respectively. *β* is the geometric factor of a spherical nucleus, 16π/3. *ʋ_m_*, which was 36.93 cm^3^·mol^−1^ for calcite, is the molar volume. *N_A_* represented Avogadro’s constant (mol^−1^). f(θ) is the correction factor. *R* is the gas constant (J·mol^−1^. K^−1^). *T* and *S* are the absolute temperature (K) and the supersaturation ratio for different solution concentrations, respectively. Each value was the average value of repeated three times.

### 3.7. Determination of Corrosion Inhibition Efficiency

The ability of PASP and PASP/5–AVA to inhibit electrochemical corrosion on carbon steel was studied at different concentrations with a CHI 660E electrochemical workstation, a three-electrode system, and the corrosion medium was 3.5% NaCl solution (298 K) [35]. The counter electrode and the reference electrode were a platinum electrode (2 cm^2^) and a saturated calomel electrode, respectively. The working electrode was 1 cm^2^ of an epoxy-clad 20^#^ carbon steel. Before the experiment began, the working electrode was polished with sandpapers of different specifications (400–2000 mesh) to ensure a smooth surface, cleaned with acetone, and dried. Before the tests, to determine the steady-state open circuit potential (*E*_OCP_), the electrode was immersed in 500 mL corrosion solution for 1 h. Potentiodynamic polarization was tested at a scan rate of 1 mv/s within the potential range ±250 mV *E*_OCP_. The metal surface coverage (*θ*) and inhibition efficiency (*η_p_*) were obtained from Tafel curves, which were calculated with Formulas (3) and (4), respectively.
(3)$$\eta_{p}\%=\frac{I_{corr}^{0}-I_{corr}}{I_{corr}^{0}}\times 100$$
(4)$$\theta =\frac{n_{p}}{100}$$

$I_{corr}^{0}$ and $I_{corr}$ are the current densities in corrosive media without and with corrosion inhibitors, respectively.

### 3.8. Mechanism Analysis

To investigate the scale inhibition mechanism at the B3LYP/6-31 G(d, p) level, the molecular structure of scale inhibitors was optimized with the Gaussian 09 software package. The structure–activity relationship was analyzed using Density Functional Theory (DFT) [36]. The highest and the lowest occupied molecular orbital energy (*E*_HOMO_ and *E*_LUMO_) were obtained. The energy gap (Δ*E*) was the difference value between *E*_HOMO_ and *E*_LUMO_ (Formula (5)). The electron transfer (Δ*N*) between *E*_HOMO_ and *E*_LUMO_ was calculated with Formulas (6) to (8).
(5)$$\Delta E=E_{LUMO}-E_{HOMO}$$
(6)$$\Delta N=\frac{\chi_{Fe}-\chi_{inh}}{2(\eta_{Fe}+\eta_{inh)}}$$
(7)$$\chi_{inh}=\frac{{-E}_{HOMO}-E_{LUMO}}{2}$$
(8)$$\eta_{inh}=\frac{E_{LOMO}-E_{HUMO}}{2}$$

*χ* is the electronegativity of the particle, *η* is the overall hardness of the substance, and in theory, the χ_Fe_ and *η*_Fe_ of pure iron were 7 eV·mol^−1^ and 0 eV·mol^−1^, respectively. Each value was the mean value, at least in triplicate.

## 4. Conclusions

The new corrosion inhibitor PASP/5–AVA was synthesized from poly(succinimide) (PSI) and 5-aminovaleric acid (5-AVA) in a mixed reaction system of organic solvent and water. The experimental results suggest that PASP/5–AVA has better scale inhibition on CaCO_3_ and CaSO_4_ and corrosion inhibition effects than PASP. Mechanistic analysis showed that PASP/5–AVA acted on CaCO_3_ through chelation solubilization and lattice distortion and on CaSO_4_ through dispersion to achieve scale inhibition. Moreover, the increasing of the -CH_2_- groups in the side chain and extending its length can enhance the scale and corrosion inhibition capability of PASP.

## Figures and Tables

**Figure 1 ijms-25-10150-f001:**
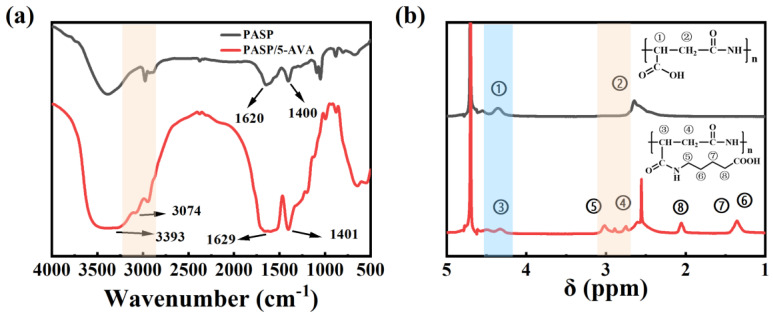
Infrared (FTIR, (**a**)) and Nuclear Magnetic Resonance spectra (^1^H-NMR, (**b**)) of PASP and PASP/5–AVA.

**Figure 2 ijms-25-10150-f002:**
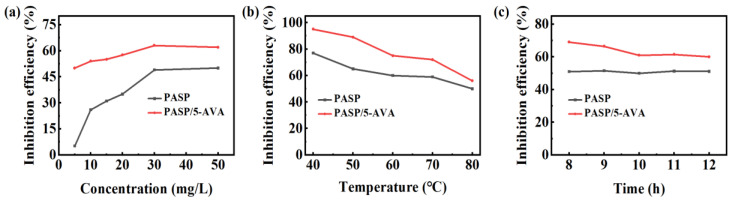
The anti-CaCO_3_ scale efficiency of PASP and PASP/5–AVA. (**a**) With different concentrations; (**b**) Different experimental temperatures (the dosage of inhibitor is 30 mg/L); (**c**) Heating time (the dosage of inhibitor is 30 mg/L).

**Figure 3 ijms-25-10150-f003:**
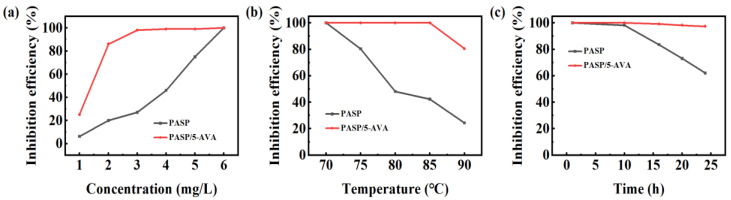
The anti-CaSO_4_ scale efficiency of PASP and PASP/5–AVA. (**a**) With different concentrations; (**b**) Different experimental temperatures (the dosage of inhibitor is 3 mg/L); (**c**) Heating time (the dosage of inhibitor is 3 mg/L).

**Figure 4 ijms-25-10150-f004:**
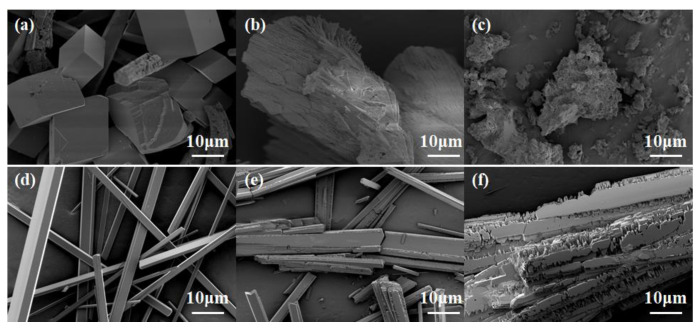
The morphology of CaCO_3_ (**a**–**c**) and CaSO_4_ (**d**–**f**) scales under SEM at different dosages. (**a**,**d**) In the absence of antiscalant; (**b**) in the presence of 30 mg/L PASP; (**c**) in the presence of 30 mg/L PASP/5–AVA; (**e**) in the presence of 3 mg/L PASP; (**f**) in the presence of 3 mg/L PASP/5–AVA.

**Figure 5 ijms-25-10150-f005:**
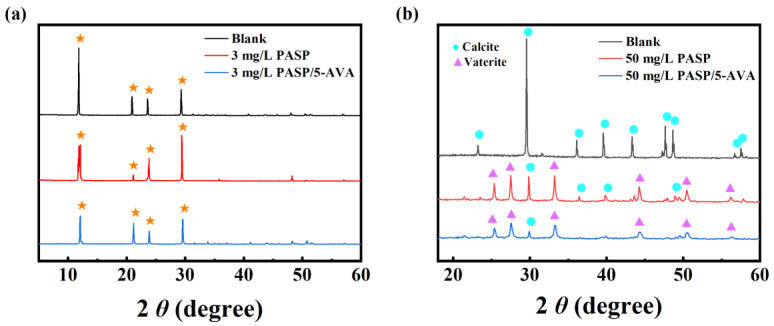
XRD patterns of CaSO_4_ (**a**) and CaCO_3_ (**b**) scales.

**Figure 6 ijms-25-10150-f006:**
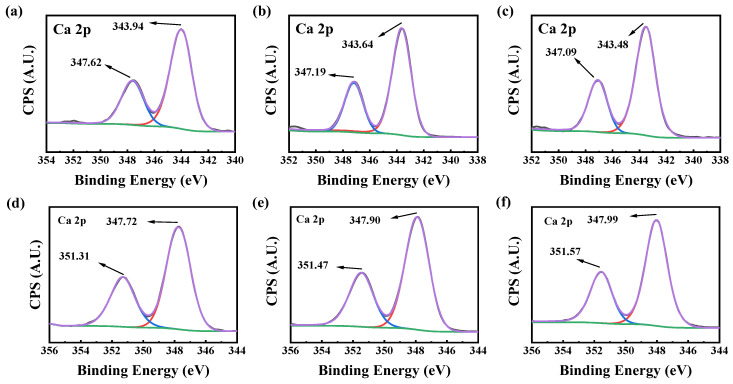
XPS spectra of Ca 2p with CaCO_3_ (**a**–**c**) and CaSO_4_ (**d**–**f**) scales. (**a**,**d**) In the absence of antiscalant; (**b**) in the presence of 30 mg/L PASP; (**c**) in the presence of 30 mg/L PASP/5–AVA; (**e**) in the presence of 3 mg/L PASP; (**f**) in the presence of 3 mg/L PASP/5–AVA.

**Figure 7 ijms-25-10150-f007:**
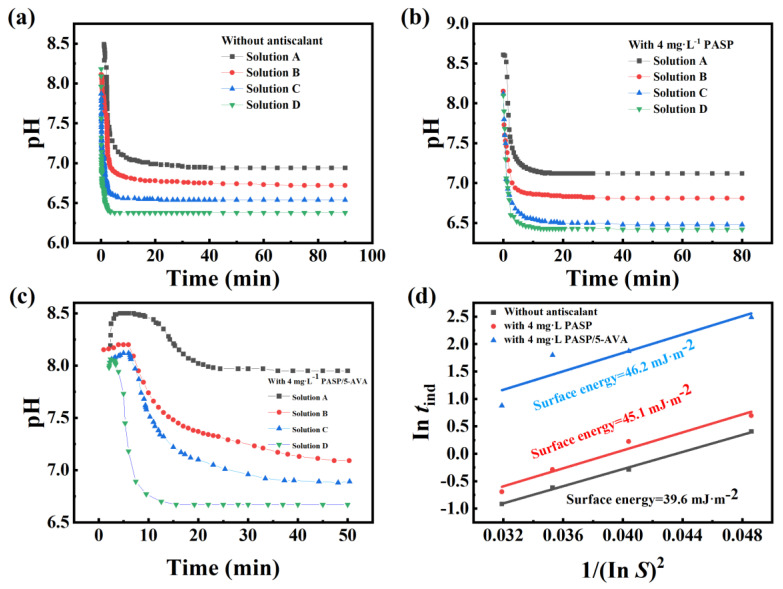
The relationship curves between pH values and CaCO_3_ scale solutions. (**a**) In the absence of antiscalant; (**b**) in the presence of 4 mg/L PASP; (**c**) in the presence of 4 mg/L PASP/5–AVA; (**d**) the relation curves between In*t*_ind_ and 1/(In*S*)^2^ with and without antiscalants.

**Figure 8 ijms-25-10150-f008:**
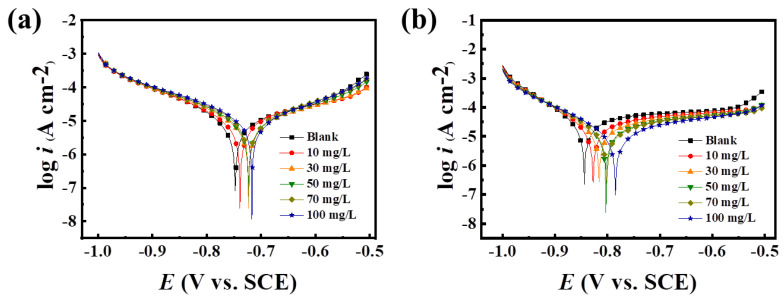
The potentiodynamic polarization curves of PASP (**a**) and PASP/5–AVA (**b**) at different concentrations in 3.5% NaCl solution.

**Figure 9 ijms-25-10150-f009:**
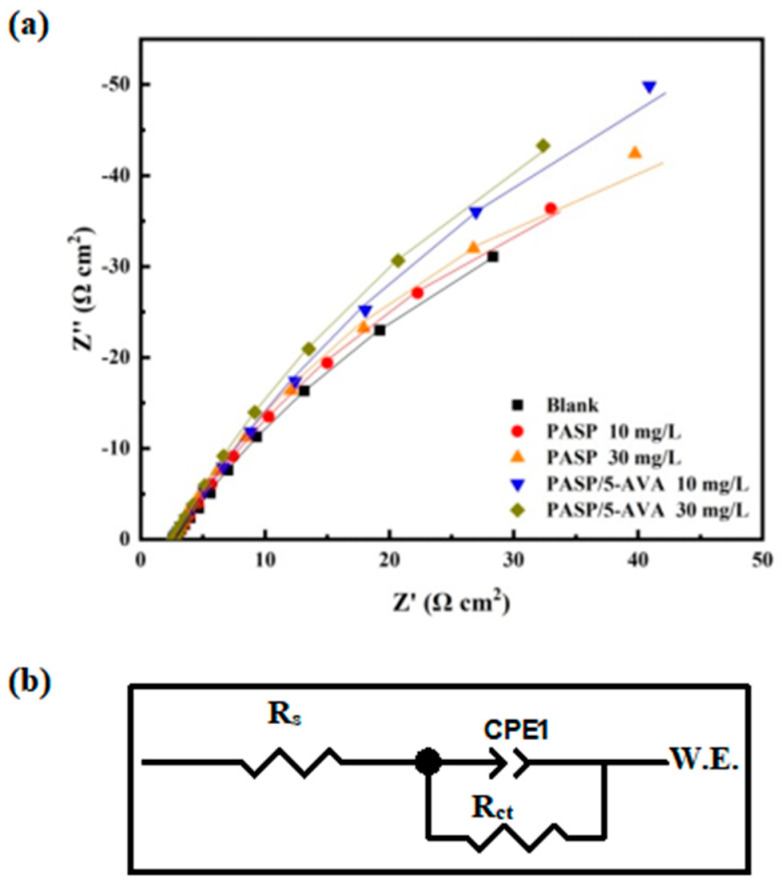
Impedance graph (**a**) for PASP and PASP/5–AVA with different concentrations and the corre-sponding analog equivalent circuit model (**b**).

**Figure 10 ijms-25-10150-f010:**
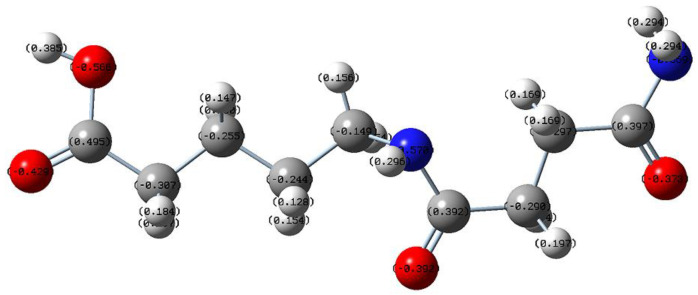
The PASP/5–AVA optimization structure diagram. The red ball (●) represents the N atom, the red ball (●) represents the O atom, the gray ball (●) represents the C atom, and the white ball (●) represents the H atom.

**Figure 11 ijms-25-10150-f011:**
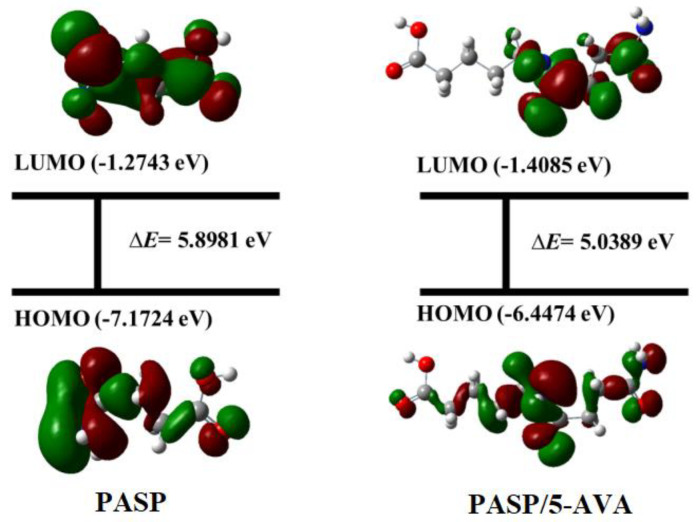
The geometrically optimized structures and molecular orbital density distribution of PASP and PASP/5–AVA according to quantum chemical calculation results.

**Figure 12 ijms-25-10150-f012:**
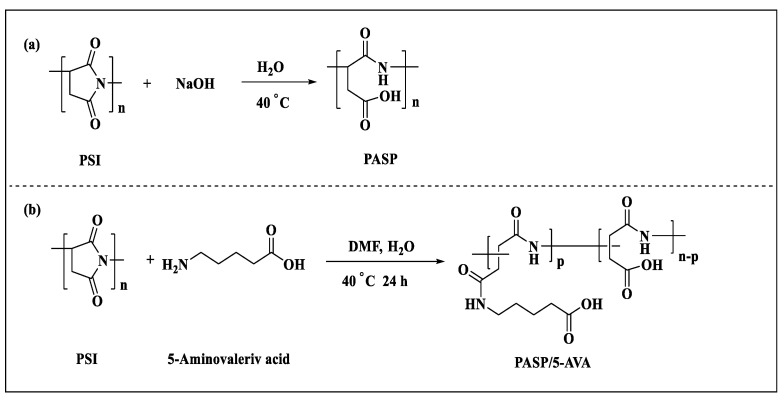
Preparation route of the PASP (**a**) and PASP/5–AVA (**b**) modified copolymer.

**Table 1 ijms-25-10150-t001:** The *M_n_*, *M_w_*, and *Ð* of PASP and PASP/5–AVA.

Sample	*M_n_*	*M_w_*	*Ð*
PASP	1778	2039	1.147
PASP/5–AVA	11,328	14,109	1.245

**Table 2 ijms-25-10150-t002:** The *t*_ind_ values of the CaCO_3_ solutions with and without antiscalants in the pH measurement process.

	Different Concentrations of CaCO_3_ Solution (mol/L)	Antiscalants		
		Blank	PASP	PASP/5–AVA
*t*_ind_ (min)	0.015	1.50	2.25	53.71
	0.02	0.75	1.25	6.00
	0.025	0.54	0.75	2.19
	0.03	0.4	0.5	1.05

**Table 3 ijms-25-10150-t003:** The electrochemical polarization parameters of carbon steel in 3.5% NaCl solution containing PASP and PASP/5–AVA with different concentrations.

Inhibitors	C (mg/L)	−*E*_corr_ (Vvs.SCE)	*β*_a_ (mV/dec)	−*β*_c_ (mV/dec)	*I*_corr_ (mA/cm^2^)	*η_p_* (%)	*θ*
**Blank**	—	0.6775	202.55	102.99	3.516	-	-
PASP	30	0.6228	267.81	109.97	3.129	10.72	0.11
	50	0.6296	250.06	104.34	2.955	15.96	0.16
	70	0.6294	278.71	100.39	2.883	18.12	0.18
	100	0.6208	259.27	104.13	2.749	21.84	0.22
PASP/5–AVA	30	0.6050	153.0	117.6	1.395	31.7	0.32
	50	0.6142	141.8	113.4	1.386	32.1	0.32
	70	0.6166	149.4	114.9	1.118	45.2	0.45
	100	0.6047	143.9	118.0	1.001	51.0	0.51

**Table 4 ijms-25-10150-t004:** Corrosion electrochemical parameters of 20^#^ carbon steel measured by EIS in 3.5% NaCl solution at 298 K without and with inhibitors at various concentrations.

Inhibitors	C (mg/L)	R_ct_ (Ohm.cm^2^)	C_dl_ (F/cm^2^)	χ^2^ (×10^−3^)	*η_p_* (%)
**Blank**	—	144.3	0.40937	0.3777	-
**PASP**	10	144.8	0.37148	0.3126	6.73
	30	145.6	0.35909	0.4240	12.28
	50	147.8	0.32399	0.4360	20.86
	70	154.5	0.31749	0.4178	22.44
	100	205.4	0.31148	0.3962	23.91
PASP/5–AVA	10	57.4	0.33137	0.2571	19.96
	30	68.52	0.29042	0.4167	29.01
	50	70.46	0.27168	0.3654	33.63
	70	74.02	0.21842	0.3600	46.64
	100	75.27	0.20722	0.39947	49.38

**Table 5 ijms-25-10150-t005:** Quantum chemical parameters of PASP and PASP/5–AVA.

Inhibitor	*E*_HOMO_ (eV)	*E*_LUMO_ (eV)	∆*E* (eV)	*χ* (eV)	*η* (eV)	∆*N*
PASP	−7.1724	−1.2743	5.8981	4.2233	2.9491	0.4708
PASP/5–AVA	−6.4474	−1.4085	5.0389	4.0379	2.6294	0.5633

**Table 6 ijms-25-10150-t006:** The reagents and equipment used in this experiment.

	Name	Purchasing Company
**Reagents**	Polysuccinimide (PSI, Mw = 7000, AR)	Wuhan Yuancheng Gongyi Technology Company Limited (Wuhan, China)
	5-Aminovaleric acid (AR)	Tianjin De’en Chemical Reagents Company Limited (Tianjin, China)
	Anhydrous sodium sulfate (AR, 99%)	Tianjin Komi Chemical Reagent Company Limited (Tianjin, China)
	Sodium carbonate (AR, 99.8%)	Tianjin Komi Chemical Reagent Co., Ltd. (Tianjin, China)
	Potassium hydroxide (AR, 90%)	Tianjin Komi Chemical Reagent Co., Ltd. (Tianjin, China)
	Anhydrous borax (AR, 95%)	Shanghai Shaoyuan Chemical Reagent Co., Ltd. (Shanghai, China)
	Calcium chloride (AR, 99.9%)	Shanghai Shaoyuan Chemical Reagent Co., Ltd. (Shanghai, China)
	Potassium chloride (AR, 99.999%)	Shanghai Shaoyuan Chemical Reagent Co., Ltd. (Shanghai, China)
	Sodium chloride (AR, 99%)	Shanghai Shaoyuan Chemical Reagent Co., Ltd. (Shanghai, China)
	Ethylenediaminetetraacetic acid disodium salt dihydrate (EDTA)	Shanghai Energy Chemical Reagent Co. Ltd. (Shanghai, China)
	Absolute ethanol	Anhui Ante Food Co., Ltd. (Anhui China)
	Ethanol absolute (AR, 99.5%)	Anhui Ante Food Co., Ltd. (Anhui, China)
	Hydrochloric acid (ω = 36%)	China Pingmei Shenma Group Kaifeng Dongda Chemical Co., Ltd. (Kaifeng, China)
Equipment	An AVANCE 400 nuclear magnetic resonance spectrometer	Bruker Co., Ltd. (Bilerika, Germany)
	An ESCALAB 250Xi X-ray photoelectron spectrometer	Thermo Fisher Scientific Co., Ltd. (waltham, USA)
	A JSM-7610F SEM	Japan Electronics Co., Ltd. (Tokyo Metropolis, Japan)
	An HH-601 constant-temperature water tank	Jintan Jingda Instrument Manufacturing Co., Ltd. (Changzhou, China)
	A Nano ZS particle size and zeta potential analyzer	Malvery Instruments Ltd. (Marvin, England)
	A PL-GPC50 gel permeation chromatograph	Agilent Technology Co., Ltd. (Palo Alto City, USA)
	A DDS-11A conductivity meter	INASE Scientific Instrument Co., Ltd. (Shanghai, China)
	A D8 Advance X-ray powder diffractometer	Bruker Co., Ltd. (Bilerika, Germany)
	A CHI660E electrochemical workstation	Chenhua Instrument Co., Ltd. (Shanghai, China)
	A VERTEX 70 Fourier transform infrared spectrometer	Bruker Co., Ltd. (Bilerika, Germany)

## Data Availability

All data are contained within the manuscript and are available upon request.

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
