# Peer review of "Synthesis and Mechanism of a Green Scale and Corrosion Inhibitor"

_ijms, 2024, doi:10.3390/ijms251810150_

Round 1

Reviewer 1 Report

Comments and Suggestions for Authors

Manuscript Number: ijms-3171299, entitled: Synthesis and mechanism of a green scale and corrosion inhibitor, presented the results of the use of poly(aspartic acid)-modified polymers (PASP/5-AVA) as the scale and corrosion inhibitors.

I believe the procedure for the synthesis of PASP/5-AVA is not acceptable. The author also did not describe the synthesis of PASP from PSI. The corrosion results are disappointing. The paper cannot be accepted for the following comments:

i)                   Line 11: You wrote: Novel……using succinimide Query/comment:  Change ‘succinimide’ to polysuccinimide & ‘novel’ to ‘new’.

ii)                 Table 1: You wrote: Mn of PASP and PASP/5-AVA) as 1778 and 11328 g mol-1, respectively.….. Query/comment: PASP is made from PSI (polysuccinimide) with an unit molar mass of 97 g mol-1 and 5-AVA with an unit molar mass of 117 g mol-1. So, the PASP/5-AVA should have a molar mass 1778*(97+117)/97 i.e. 3922 g mol-1. Justify: why is it 11328 g mol-1?

iii)               Fig. 2: You wrote: (b) Different experimental temperature; (c) Heating time: Query/comment: In (b) and (c) you must mention the concentration (ppm) of the antiscalants.

iv)               Tables 3 & 4: Query/comment: The low inhibition efficiency for both the inhibitors revealed the inability to impart corrosion inhibition on mild steel.

v)                  Line 513----:  Query/comment: 0.984 g polysuccinimide (PSI) and 0.231 g of 5-aminovaleric acid are not stoichiometric (i.e. 1:1 mole ratio in repeating units). There is PSI in plenty of excess and can not generate the fully transformed form as depicted in Fig. 12.

vi)               Line 521: You wrote: and then spun to obtain the PASP/5-AVA solid product. Query/comment: How much product is obtained? Indicate the percent yield. 

Comments on the Quality of English Language

In some places 'typos'.

Author Response

Dear reviewer,

Thank you very much for your hard work in reviewing this manuscript and even more for the valuable suggestions you have given us. In response to your suggestions, we have revised and amended them one by one. We hope you will give us a chance. The explanations to some of your questions are listed below for your review. If you have any questions, we hope you can give us feedback, and we will make serious changes and revisions again. Thank you very much.

  • I believe the procedure for the synthesis of PASP/5-AVA is not acceptable. The author also did not describe the synthesis of PASP from PSI.

Response: Honorable Reviewer, Thank you very much for your valuable suggestions. The polyaspartic acid (PASP) used in this experiment was prepared by the classical alkaline hydrolysis of polysuccinimide (PSI), which has been reported in much related literature, so please forgive us for our carelessness in not writing this step clearly. We have included the method and reaction route for the preparation of PASP in the manuscript. Please review.

Moreover, to increase the yield of the target product PASP/5-AVA, we tried to change various reaction conditions, including the reaction solvent. Finally, the target product was successfully prepared in high yield following the steps described in the manuscript and its structure was verified by infrared NMR spectroscopy. I am deeply sorry for the misunderstanding caused by our inadvertence in drawing the reaction equation with 100 per cent grafting. Also, thank you again for your seriousness and the feedback you gave me. We have revised the reaction equation. Please review.

  1. Line 11:You wrote: Novel……using succinimide Query/comment:  Change ‘succinimide’ to polysuccinimide & ‘novel’ to ‘new’.

Response: Honorable Reviewer, Thank you very much for your valuable suggestions. We have changed "novel" to "new" in the manuscript. Please review.

  1. ii)  Table 1: You wrote:Mnof PASP and PASP/5-AVA) as 1778 and 11328 g

mol-1, respectively.….. Query/comment: PASP is made from PSI (polysuccinimide) with an unit molar mass of 97 g mol-1 and 5-AVA with an unit molar mass of 117 g mol-1. So, the PASP/5-AVA should have a molar mass 1778*(97+117)/97 i.e. 3922 g mol-1. Justify: why is it 11328 g mol-1?

Response: Honorable Reviewer, I agree with your proposal very much and thank you for your valuable questions. On this issue, we have the same question when this result was analyzed with GPC. Our analysis speculates that the PASP/5-AVA polymer is not composed of a single chain segment, but there are longer chain segments, in other words, there are multiple chain segments with different molecular weights in the PASP/5-AVA polymer, and therefore the actual Mn in the GPC analysis result is much larger than the theoretical value. To reduce readers' doubts, we add this explanation to the “ 2.1.2. Gel chromatography (GPC) analysis”. Please review.

iii) Fig. 2: You wrote: (b) Different experimental temperature; (c) Heating time: Query/comment: In (b) and (c) you must mention the concentration (ppm) of the antiscalants.

Response: Honorable Reviewer, thank you very much for your careful and responsible review of the manuscript and your valuable suggestions. We've added the scale inhibitor concentration in the Figure 2 and Figure 3. At the same time, Lines 169 and 192 of the manuscript indicate the concentration of inhibitor is 30mg/L in Figure 2(b) Different experimental temperature and Figure 2(c) Heating time, respectively. Lines 222 and 238 indicate that the concentration of the agent is 3mg/L in Figure 3(b) and 3(c), respectively. Please review.

  1. iv)Tables 3 & 4Query/comment: The low inhibition efficiency for both the inhibitors revealed the inability to impart corrosion inhibition on mild steel.

Response: Honorable Reviewer, Thank you very much for your valuable questions. As a new type of green water treatment agent, the advantages of polyaspartic acid (PASP) are mainly reflected in scale inhibition, especially calcium sulphate scale inhibition, but the corrosion inhibition performance has been less than ideal and difficult to improve. Compared with PASP, the corrosion inhibition efficiency of PASP/5-AVA has been improved up to nearly 30%, which is already considered a great improvement in relative terms. Therefore, in this paper, although the scale inhibition performance is mainly studied, it is also proposed to improve its corrosion inhibition performance, although it does not reach the expected value. This provides a theoretical reference for the subsequent modification of PASP to improve its corrosion inhibition performance. 

  1. v)Line 513----:  Query/comment:984 g polysuccinimide (PSI) and 0.231 g of 5-aminovaleric acid are not stoichiometric (i.e. 1:1 mole ratio in repeating units). Response:Honorable Reviewer, thank you very much for your serious and responsible attitude and for pointing out our errors in the writing process. After reading the trial notes, we found that 0.231g was our initial trial data. The final dose we used was 1.171 g 5-aminovaleric acid. The amount of substance (n) of both PSI and 5-AVA was 10 mM. Please forgive us for such a vulgar mistake. We've made corrections in the manuscript. Please review.

  1. vi)Line 521:You wrote: and then spun to obtain the PASP/5-AVA solid product. Query/comment: How much product is obtained? Indicate the percent yield. 

Response: Honorable Reviewer, thank you very much for your valuable advice. Firstly, we are very sorry to have missed this data. After reviewing our experimental records, the yield of PASP/5-AVA under these reaction conditions is quite promising, up to about 96%. We have supplemented the yield of PASP/5-AVA in 3.2. Synthesis of PASP/5-AVA. Please review.

Reviewer 2 Report

Comments and Suggestions for Authors

Review Comments

Manuscript: “Synthesis and mechanism of a green scale and corrosion inhibitor.”

General Comments

i.               The manuscript reports a study on novel poly(aspartic acid)-modified polymers (PASP/5-AVA) as a green water treatment agent. The proposed modified PASP/5-AVA structure was characterized in terms of corrosion inhibition and mineral scale retardation over range of temperature and concentration. The authors also explored how the modified antiscalant retard corrosion and mineral nucleation via XRD, XPS, SEM and quantum chemistry analysis. The authors claimed that PASP/5-AVA exhibited better scale and corrosion inhibition compared to PASP and maintained efficacy and thermal stability of scale inhibition effect for a long time. 

ii.             Based on the result presented by the authors, source water chemistry and location where CaCO3 and CaSO4 scaling propensity is found is not discussed in the manuscript. The degree of supersaturation with respect to mineral scalant of concern is also not provided. At what water treatment approach would one expect to use such high temperature that the proposed antiscalant agent will need to have some thermal stability? Antiscalants are expensive, what will be the tradeoff between scaling and corrosion inhibition compared to increase in overall water treatment cost. Corrosion inhibition and related problems to water treatment is also not discussed in the introduction of the proposed manuscript. What is the relation between scaling problems which are known problems in water treatment and corrosion? Perhaps the authors should elaborate at least in the introduction.

Specific comments that illustrate the concerns with the manuscript, and provide suggestions for improving the study, are provided below with the intent of assisting the authors to improve their study:

1. Introduction, Lines 57-58. The authors made a statement that “However, thus far, the structure-activity relationships of the water treatment agent are still not clear and lack of strong scientific evidence”. This is too abstract and misleading, there are many studies who reported how PASP modified antiscalant retard both CaCO3 and CaSO4 in both static test and in RO tests from mechanistic point of view (from google scholar search). The authors are therefore advice those papers before coming to such a conclusion. 

2. Introduction lines. The authors state that “but also haver disadvantages such as large dosages.” There seems to be a typo, could the sentence be “but also have a disadvantages such as high dosages

3. Introduction, lines 41-45. The authors stated that “Some phosphorus-containing scale inhibitors, for example, organic phosphonates, have better scale inhibition efficiencies, however, they cause eutrophication of water bodies, rapid growth of algae and plankton, which hinder the widespread popularization of phosphorus-containing scale inhibitors [10,11]” is misleading as “eutrophication of water bodies, rapid growth of algae and plankton” means the same thing. Eutrophication means excessive nutrients accumulate in a body of water, resulting in an increased growth of microorganisms such as algae, plankton etc. Furthermore, there seems to be mis citation of references [10,11] as the cited papers did not make such statement as the authors claimed.

4. Introduction, Lines 57-58. The authors made a statement that “However, thus far, the structure-activity relationships of the water treatment agent are still not clear and lack of strong scientific evidence”. This is too abstract and misleading, there are many studies who reported how PASP modified antiscalant retard both CaCO3 and CaSO4 in both static test and in RO desalination tests from mechanistic point of view (from simple google scholar search). The authors are therefore advice go over those papers before coming to such a conclusion. 

5. Results and Discussion, Line 242, “When the concentration of PASP was30 mg/L”. there should be space between ‘was’ and ’30 mg/L’.

6. Results and Discussion, section 2.2. Why was the modified antiscalant not effective in retarding Calcite scaling at lower concentration compared to CaSO4? What was the basis of selecting the antiscalant initial concentration? What is the scope of using such antiscalant? Is it only applicable at high temperature application? The authors should also take into account cost of the antiscalant if it is to be applied at such high concentration for example to control calcite-based scaling. 

7. The authors should note that relationships between surface conditions, crystallization induction time, and the rate of nucleation are governed by the level of solution supersaturation, and the space time/convective residence time in the systems. The saturation indices (feed, and both for concentrate bulk solution and at the surface) should be reported for all the mineral scalant solutions utilized in the study.

8. Material and methods, section 3.4, what was the rational of selecting tests solution’s Ca2+ (240 mg/L) and HCO3(732 mg/L) for CaCO3 and Ca2+ (2000 mg/L) and SO42- for CaSO4? There seems to be typo in ‘HCO3-’ (Line 538), it should be ‘HCO3-’. What was the saturation indexes of both CaCO3 and CaSO4 as this will determine if scaling with respect to the scalants will occur or not. 

9. Materials and Methods. The experimental system as shown by the authors provides for static systems. This means that any crystals formed in the bulk of the solution, as well as crystals that detach from the surface, would recirculate through the system, and could grow, breakup and trigger the formation of other crystals, and redeposit onto the surface. This scenario is quite different from that which occurs in actual plants in which the convective residence time is orders of magnitude shorter than the residence time in static stream. Therefore, the conclusions regarding the scaling of different antiscalant and surfaces are specific to the type of experiments carried out in this study and thus may not be representative of what happens in actual systems. 

Author Response

Dear reviewer,

Sincerely thank you for reviewing this manuscript and giving us valuable suggestions. According to your suggestions, we have made changes and additions one by one. As shown below, we have marked them with red in the article for your review . If you have any questions, we hope you can give us your feedback, and we will revise and amend it again. Thank you very much!

  1. The manuscript reports a study on novel poly(aspartic acid)-modified polymers (PASP/5-AVA) as a green water treatment agent. The proposed modified PASP/5-AVA structure was characterized in terms of corrosion inhibition and mineral scale retardation over range of temperature and concentration. The authors also explored how the modified antiscalant retard corrosion and mineral nucleation via XRD, XPS, SEM and quantum chemistry analysis. The authors claimed that PASP/5-AVA exhibited better scale and corrosion inhibition compared to PASP and maintained efficacy and thermal stability of scale inhibition effect for a long time. 

Response: Honorable Reviewer, thank you again for taking the time to review our manuscript and for giving us such high marks.

  1. Based on the result presented by the authors, source water chemistry and location where CaCO3 and CaSO4 scaling propensity is found is not discussed in the manuscript. The degree of supersaturation with respect to mineral scalant of concern is also not provided. At what water treatment approach would one expect to use such high temperature that the proposed antiscalant agent will need to have some thermal stability? Antiscalants are expensive, what will be the tradeoff between scaling and corrosion inhibition compared to increase in overall water treatment cost. Corrosion inhibition and related problems to water treatment is also not discussed in the introduction of the proposed manuscript. What is the relation between scaling problems which are known problems in water treatment and corrosion? Perhaps the authors should elaborate at least in the introduction.

Response: Honorable Reviewer, thank you for such valuable advice! In the production process of industry and other businesses, 90% of the production process is exothermic, which, if not eliminated in time, will not only cause quality problems in the production of products, but also lead to serious accidents in the factory, such as explosions. For this reason, circulating cooling water systems are commonly used in industrial businesses and the like to ensure that excess heat is removed in a timely manner. The temperature of industrial circulating cooling water is usually required to be maintained at 15℃ to 45℃, and fluctuates as the system operates, with a gradual increase in temperature. With the long-term operation of the system, water will continue to evaporate, the ion concentration in the water, etc. will gradually concentrate and increase, it is easier to combine with inorganic salt ions to form calcium scale deposits. Scale deposition on the surfaces of equipment and pipelines, particularly around pipe bends, block the equipment and pipelines, reduce the heat transfer efficiency, and cause heat accumulation, which result in accidents. In addition, scale deposits cause localised current differentials in pipelines, accelerating corrosion of equipment and pipelines. Therefore, it is necessary to add water treatment agents with scale and corrosion inhibiting properties and a relatively wide temperature range to reduce scale deposition in industrial circulating cooling water, protect metal surfaces and reduce corrosion. We have refined this section in the first paragraph of the introduction. Please review.

In addition, the polyaspartic acid series (PASPs) of water treatment agents have become one of the most promising green water treatment agents due to their unique biodegradable structure, but due to a variety of factors, these water treatment agents are not widely used. This manuscript focuses on the water treatment performance of the newly prepared PASP/5-AVA, and the influence of functional groups on the scale and corrosion inhibition performance of PASP, to provide a little theoretical guidance for the design of the future PASP series of water treatment agents, so we do not consider the cost of its wide application at present.

  1. Introduction, Lines 57-58. The authors made a statement that “However, thus far, the structure-activity relationships of the water treatment agent are still not clear and lack of strong scientific evidence”. This is too abstract and misleading, there are many studies who reported how PASP modified antiscalant retard both CaCO3and CaSOin both static test and in RO tests from mechanistic point of view (from google scholar search). The authors are therefore advice those papers before coming to such a conclusion. 

Response: Honorable Reviewer, forgive the inaccuracy of our expression, we think you are absolutely right. We have already changed this sentence to “Although much has been reported about the structure-activity relationship of water treatment agents, more and deeper scientific evidence is needed to confirm it ”.

  1. Introduction lines. The authors state that “but also haver disadvantages such as large dosages.” There seems to be a typo, could the sentence be “but also have a disadvantages such as high dosages.

Response: Honorable Reviewer, thank you very much for your valuable advice. We have made corrections in the manuscript, please forgive us for such a vulgar error.

  1. Introduction, lines 41-45. The authors stated that “Some phosphorus-containing scale inhibitors, for example, organic phosphonates, have better scale inhibition efficiencies, however, they cause eutrophication of water bodies, rapid growth of algae and plankton, which hinder the widespread popularization of phosphorus-containing scale inhibitors [10,11]” is misleading as “eutrophication of water bodies, rapid growth of algae and plankton” means the same thing. Eutrophication means excessive nutrients accumulate in a body of water, resulting in an increased growth of microorganisms such as algae, plankton etc. Furthermore, there seems to be mis citation of references [10,11] as the cited papers did not make such statement as the authors claimed.

Response: Honorable Reviewer, thank you very much for reviewing the manuscripts so carefully and responsibly. We have carefully changed this reference. Please review.

“10. Hu, Y.L.; Chen, C.M.; Liu, S.T. State of art bio-materials as scale inhibitors in recirculating cooling water system: a review article. Water Sci. Technol. 2022, 85 (5), 1500−1521. DOI: 10.2166/wst.2022.027.

  1. Yi, X.Y. ; Yang, S.; He, X.; Wang, Z.W.;Rui, M.; Tang, Y.L. Use of modified poly-epoxysuccinic acid as an efficient scale inhibitor to control CaSO4 scaling in NF processes: Performance and mechanisms. Desalination 2024, 586, 117821. https://doi.org/10.1016/j.desal.2024.117821.”

  1. Introduction, Lines 57-58. The authors made a statement that “However, thus far, the structure-activity relationships of the water treatment agent are still not clear and lack of strong scientific evidence”. This is too abstract and misleading, there are many studies who reported how PASP modified antiscalant retard both CaCO3and CaSO4 in both static test and in RO desalination tests from mechanistic point of view (from simple google scholar search). The authors are therefore advice go over those papers before coming to such a conclusion. 

Response: Honorable Reviewer, thank you for your valuable advice. We have rewritten this sentence in the manuscript.

  1. Results and Discussion, Line 242, “When the concentration of PASP was30 mg/L”. there should be space between ‘was’ and ’30 mg/L’.

Response: Honorable Reviewer, thank you for your valuable advice. We have revised it in the manuscript.

  1. Results and Discussion, section 2.2. Why was the modified antiscalant not effective in retarding Calcite scaling at lower concentration compared to CaSO4? What was the basis of selecting the antiscalant initial concentration? What is the scope of using such antiscalant? Is it only applicable at high temperature application? The authors should also take into account cost of the antiscalant if it is to be applied at such high concentration for example to control calcite-based scaling. 

Response: Honorable Reviewer, thank you for your valuable advice. Different types of scale inhibitors for different types of calcium scale, the inhibition effect may not be the same, which has a lot to do with the scale inhibitor and different types of calcium scale and other factors such as the mechanism of action. Mechanistic analysis showed that PASP/5-AVA acted on CaCO3 through chelation solubilisation and lattice distortion, and on CaSO4 through dispersion to achieve scale inhibition. In addition, the solubility of calcium sulphate and calcium carbonate, as well as the process of calcium scale crystal generation are not the same, so the same scale inhibitor will eventually show different inhibition concentration or effect. For example, organophosphate scale inhibitors inhibit calcium sulfate scale extremely well, but inhibit calcium carbonate poorly.

The scale inhibitors used in actual circulating cooling water systems are usually compounded. Polyaspartic acid (PASP) is considered to be one of the promising green water treatment agents due to its unique degradation structure, but it has not really been applied to actual circulating water and is still at the research and exploration stage. Therefore, its cost is not currently considered in the synthesis process. Secondly, the initial study concentrations of calcium sulphate scale inhibitor and calcium carbonate scale inhibitor are empirical values from our laboratory as well as other peer studies (https://doi.org/10.1016/j.desal.2022.116101 ). 

The applicable range of scale inhibitor should be determined according to the actual application of compounding, if a single scale inhibitor, usually choose the highest rate of scale inhibition when the concentration of the agent, in order to save costs, while reducing the environmental impact and so on. For example, in this experiment, at 30 mg / L, scale inhibitor inhibition of calcium carbonate scale is the best effect, so it will choose 30 mg / L. The temperature of circulating water in chemical plants is usually controlled between 15 °C and 45 °C, but it fluctuates and is usually on the high side due to a number of factors. Therefore, on the premise of ensuring that scale inhibitors maintain high scale inhibition efficiency in the commonly used temperature range, the scale inhibition effect of scale inhibitors at higher temperatures was investigated for reference.

  1. The authors should note that relationships between surface conditions, crystallization induction time, and the rate of nucleation are governed by the level of solution supersaturation, and the space time/convective residence time in the systems. The saturation indices (feed, and both for concentrate bulk solution and at the surface) should be reported for all the mineral scalant solutions utilized in the study.

Response: Honorable Reviewer, thank you for your valuable advice.We think your proposal is very meaningful and worthy of our in-depth discussion and research. However, our current manuscript focuses on exploring the scale inhibition effect of the new scale inhibitor PASP/5-AVA as well as its scale inhibition mechanism, all of which are currently being tested according to the national standard for determining the scale inhibition properties of water treatment agents in China (GB/T 16632-2019). As a next step, based on your suggestion, we will fully investigate the effect of its saturation index on the scale inhibition efficiency of the scale inhibitor and possibly publish it in the next new manuscript.

  1. Material and methods, section 3.4, what was the rational of selecting tests solution’s Ca2+ (240 mg/L) and HCO3- (732 mg/L) for CaCO3 and Ca2+ (2000 mg/L) and SO42- for CaSO4? There seems to be typo in ‘HCO3-’ (Line 538), it should be ‘HCO3-’. What was the saturation indexes of both CaCO3 and CaSO4 as this will determine if scaling with respect to the scalants will occur or not. 

Response: Honorable Reviewer, thank you very much for your comments and suggestions. In this experiment, the scale inhibition effect of scale inhibitor on CaCO3 scale and CaSO4 scale is tested according to the Chinese national standard (GB/T 16632-2019) and Chinese industry standard (QSY126-2014) respectively, and these concentrations are not decided by ourselves. ‘HCO3-’ (Line 538 (Line 576)) has been revised.

The pH of the calcium scale solution system configured in accordance with national standards is 8.8 to 9. As can be seen from the test results in Fig. 7a, calcium carbonate tends to precipitate and is about to be saturated at a pH between 6.5 and 8.5. This indicates that the calcium scale system configured in the national standard is inclined to precipitate calcium scale within the experimental range. Likewise with calcium sulphate.

  1. Materials and Methods. The experimental system as shown by the authors provides for static systems. This means that any crystals formed in the bulk of the solution, as well as crystals that detach from the surface, would recirculate through the system, and could grow, breakup and trigger the formation of other crystals, and redeposit onto the surface. This scenario is quite different from that which occurs in actual plants in which the convective residence time is orders of magnitude shorter than the residence time in static stream. Therefore, the conclusions regarding the scaling of different antiscalant and surfaces are specific to the type of experiments carried out in this study and thus may not be representative of what happens in actual systems. 

Response: Honorable Reviewer, thank you very much for your suggestions. We strongly agree with this. However, at the moment, our national test for scale inhibition of scale inhibitors in industrial circulating cooling water is explicitly based on static scale inhibition test methods as well as industry standard. We do take this into consideration and therefore in our future research we will develop new methods to test the scale inhibition of scale inhibitors in close proximity to water flow conditions.
